# R-Spondin2, a Positive Canonical WNT Signaling Regulator, Controls the Expansion and Differentiation of Distal Lung Epithelial Stem/Progenitor Cells in Mice

**DOI:** 10.3390/ijms23063089

**Published:** 2022-03-13

**Authors:** Ahmed A. Raslan, Youn Jeong Oh, Yong Ri Jin, Jeong Kyo Yoon

**Affiliations:** 1Department of Integrated Biomedical Science, Graduate School, Soonchunhyang University, 25 Bongjeong-ro, Dongnam-gu, Cheonan 31151, Korea; ahmed.raslan@sch.ac.kr; 2Soonchunhyang Institute of Medi-Bio Science, Soonchunhyang University, 25 Bongjeong-ro, Dongnam-gu, Cheonan 31151, Korea; yjoh@sch.ac.kr; 3Center for Molecular Medicine, Maine Medical Center Research Institute, 81 Research Drive, Scarborough, ME 04074, USA; jiny@mmc.org

**Keywords:** AT2 cells, β-catenin, club cells, lung homeostasis, lung regeneration, lung stem/progenitor cells, *R-spondin*, RSPO2, WNT signaling

## Abstract

The lungs have a remarkable ability to regenerate damaged tissues caused by acute injury. Many lung diseases, especially chronic lung diseases, are associated with a reduced or disrupted regeneration potential of the lungs. Therefore, understanding the underlying mechanisms of the regenerative capacity of the lungs offers the potential to identify novel therapeutic targets for these diseases. R-spondin2, a co-activator of WNT/β-catenin signaling, plays an important role in embryonic murine lung development. However, the role of *Rspo2* in adult lung homeostasis and regeneration remains unknown. The aim of this study is to determine *Rspo2* function in distal lung stem/progenitor cells and adult lung regeneration. In this study, we found that robust *Rspo2* expression was detected in different epithelial cells, including airway club cells and alveolar type 2 (AT2) cells in the adult lungs. However, *Rspo2* expression significantly decreased during the first week after naphthalene-induced airway injury and was restored by day 14 post-injury. In ex vivo 3D organoid culture, recombinant RSPO2 promoted the colony formation and differentiation of both club and AT2 cells through the activation of canonical WNT signaling. In contrast, *Rspo2* ablation in club and AT2 cells significantly disrupted their expansion capacity in the ex vivo 3D organoid culture. Furthermore, mice lacking *Rspo2* showed significant defects in airway regeneration after naphthalene-induced injury. Our results strongly suggest that RSPO2 plays a key role in the adult lung epithelial stem/progenitor cells during homeostasis and regeneration, and therefore, it may be a potential therapeutic target for chronic lung diseases with reduced regenerative capability.

## 1. Introduction

The lungs are essential organs for gas exchange between the blood and the external atmosphere as well as for immune defense against external pathogens and environmental factors. The lungs need to maintain their cellular composition and three-dimensional structure throughout an organism’s lifetime to perform their functions. Although the lungs exhibit a relatively low rate of cellular turnover under normal, healthy conditions, they show a remarkable regenerative capacity post-injury or damage that largely depends on the resident stem/progenitor cells [1,2,3,4]. The histological complexity and distinct functional features of each region of the lungs are well reflected in the identities of specific resident stem/progenitors [1,2,3,4].

Several cell populations in the trachea, including cytokeratin5 (KRT5^+ve^) basal cells (BCs), submucosal gland myoepithelial cells (KRT5^+ve^, KRT14^+ve^, actinα2^+ve^), and secretoglobin 1A1 (SCGB1A1^+ve)^ club cells, have been identified as distinct stem/progenitor cells [1,5,6,7]. In the bronchioles of naphthalene-induced injury model mice, variant club cells located near the neuroendocrine bodies function as stem/progenitor cells [1,3,8]. They can self-renew and differentiate into multiple epithelial lineages, including basal, ciliated, and goblet cells of the airways, and even alveoli type 1 (AT1) and surfactant protein C (SFTPC^+ve^) type 2 (AT2) cells [1,6,9,10,11,12]. Within the terminal airways close to the alveoli, a unique population of bronchioalveolar stem cells (SCGB1A1^+ve^, SFTPC^+ve^) in the bronchioalveolar duct junction can be expanded and differentiated into airway and alveolar cell types in vivo with stem/progenitor cell function [1,13,14,15]. In the alveoli, AT2 cells are the main stem/progenitor cells that can self-renew and differentiate into AT1 cells [1,13,16]. The regenerative function of these region-specific stem/progenitor cells is tightly regulated by the cross-talk between them and their microenvironment or neighboring cells [17,18,19,20].

WNT signaling is a crucial pathway for the self-renewal and specification of stem cells in multiple organs [21]. WNT signaling has been shown to be important for prenatal lung development as well as the homeostasis and regeneration of the adult lungs [1,22,23]. In the adult lungs, WNT signaling is activated in both the airways and alveolar epithelium in response to injury [1]. After naphthalene-induced lung injury in mice, the surviving club cells in the bronchioles secrete WNT ligands, including WNT3A, WNT5A, and WNT7B. These WNT ligands appear to induce the secretion of FGF10, which is a mitogenic and surviving factor for club cells, from airway smooth muscle cells (ASMCs) [17,24,25]. Thus, WNT signaling plays an important role in the regeneration of the bronchiolar epithelium. AT2 cells reside in a WNT-rich environment, and AXIN2^+ve^ AT2 (WNT-responsive) cells represent evolutionarily conserved alveolar epithelial stem/progenitor cells [26,27]. Following injury caused by bleomycin, WNT signaling is very important for the proliferation and differentiation of AT2 cells [1].

The R-spondin (RSPO) protein family can potentiate canonical WNT signaling in many cellular contexts. All four members of the RSPO protein family share a secondary structure: a signal peptide at the N-terminus, two adjacent furin-like cysteine-rich domains (FU-CRD1 and 2) followed by a single TSR-1 domain, and a positively charged basic region (BR) domain of varying lengths at the C-terminus [28,29,30,31]. In LGR4/5/6-dependent canonical WNT signaling potentiation, RSPO binds to zinc and ring finger 3 (ZNFR3)/ring finger protein 43 (RNF43) and leucine-rich repeat containing G protein-coupled receptors (LGR4/5/6) through FU-CRD1 and FU-CRD2, respectively [31,32,33]. This binding prevents the ZNRF3/RNF43-mediated degradation and membrane clearance of the WNT receptors frizzled (FZD) and low-density lipoprotein receptor-related proteins (LRP5/6). As a result, the WNT receptor complex accumulates on the plasma membrane of cells by RSPO action, resulting in the sensitization of cells to WNT ligands and the accumulation of β-catenin [31,32,33]. Unlike RSPO1/4, RSPO2/3 can potentiate canonical WNT signaling in an LGR4/5/6-independent manner [31,34,35]. In addition to the potentiation of canonical WNT signaling, RSPOs also activate the noncanonical WNT5A/PCP signaling by binding of the TSR-1 domain to heparan sulfate proteoglycans [31,36,37]. Thus, the importance of RSPOs in the regulation of canonical WNT signaling is significant and should be considered as a key canonical WNT signaling factors.

It was previously demonstrated that *Rspo2* is expressed in the lung bud mesenchyme during mouse development [38,39]. *Rspo2* knockout mice die after birth, which is likely due to lung hypoplasia and defects in lung branching [38]. The hypoplastic lungs showed a 50% reduction in weight but no change in the ratio of differentiated pulmonary epithelial cells, suggesting that hypoplasia is reflective of reduced proliferation but not failed differentiation [38]. However, the expression and physiological relevance of *Rspo2* in adult lungs are still unclear. Recent data strongly suggest RSPO2 function in the adult lungs. For instance, *Rspo2* has been shown to play a role in idiopathic lung fibrosis and is a potential biomarker for lung cancer in humans [40,41]. Recently, it has been demonstrated that RSPO2 restricts neutrophil migration from the circulation into alveolar spaces associated with increased lung permeability and/or barrier disruption in adult mouse lungs, suggesting that RSPO2 may be essential for blood–gas barrier integrity [42]. However, whether RSPO2 is expressed and plays any role in the expansion and differentiation of adult lung epithelial stem/progenitor cells during homeostasis and regeneration has not been demonstrated.

In this study, we aimed to determine the role of *Rspo2* in lung stem/progenitor cells and regeneration by utilizing *ex vivo* ALI-organoid culture using primary stem/progenitor cells, such as DLESPs, club cells, or AT2 cells and in vivo naphthalene-induced airway injury in mice lacking *Rspo2*. Our present study shows a previously unknown role of RSPO2 as a canonical WNT signaling regulator in adult lung epithelial stem/progenitor cells and airway regeneration.

## 2. Results

### 2.1. Rspo2 Is Expressed in Bronchial and Alveolar Epithelium

As a first step toward investigating the RSPO2 function in lung homeostasis and regeneration in adult mice, we determined *Rspo2* RNA expression in the adult mouse lungs by in situ hybridization analysis. We found that *Rspo2* was expressed in the bronchial epithelium and alveoli (Figure 1A). We next isolated EpCAM^+ve^ epithelial cells (also called DLESPs) from the distal lung. A major population of these isolated cells comprise AT1/AT2 cells, and other bronchial epithelial cells, such as club cells, are also included. qRT-PCR analysis using different lung epithelial and mesenchymal marker genes confirmed the efficacy of isolation (Appendix A). We determined RSPO2 protein expression in freshly isolated EpCAM^+ve^ and EpCAM^−ve^ cells by Western blotting and found that RSPO2 protein was exclusively expressed in EpCAM^+ve^ cells (Figure 1B). Immunofluorescence staining revealed RSPO2 expression in the majority of EpCAM^+ve^ cells (Figure 1C,D). Most RSPO2^+ve^ cells were SFTPC^+ve^, indicating that RSPO2 was mainly expressed in AT2 cells among the isolated EpCAM^+ve^ cells. When the isolated club cells and AT2 cells were stained, we detected both club cells and AT2 cells expressing RSPO2 protein (Figure 1E).

We further determined RSPO2 expression in adult lung tissues by immunofluorescence staining analysis using antibodies against RSPO2, along with antibodies against club cell marker, SCGB1A1, and AT2 cell marker, SFTPC. RSPO2 expression was colocalized with the expression of both markers, indicating that both club and AT2 cells express RSPO2 (Figure 1F). Additionally, we examined β-galactosidase expression in *Rspo2^LacZ/+^* mice in which the *LacZ* gene was inserted into the *Rspo2* locus to faithfully recapitulate endogenous *Rspo2* expression [43]. We confirmed robust β-galactosidase expression in both club and AT2 cells (Appendix A). Taken together, these results suggest that RSPO2 may play a critical role in the bronchial and alveolar epithelial cells.

### 2.2. RSPO2 Is a Key Autocrine Factor Required for the Colony Formation of DLESPs

To investigate the RSPO2 function in lung epithelial cells, we used ex vivo air–liquid interphase (ALI) 3D organoid culture using EpCAM^+ve^ DLESPs isolated from adult mouse lungs (Figure 2A). We treated DLESPs with recombinant RSPO2 protein for two weeks. RSPO2 treatment significantly increased the colony-forming efficiency of DLESPs (Figure 2B,C). Since RSPO2 activity is shown to be dependent on the presence of WNT ligands, we determined whether RSPO2’s role in colony formation of DLESPs depends on endogenously secreted WNTs from DLESPs and/or co-cultured fibroblasts by co-treating with WNT secretion blocker, IWP-2. We found that IWP-2 severely reversed the colony-forming activity of RSPO2, suggesting that RSPO2 enhances colony formation in a WNT-dependent manner (Figure 2D,E). In addition, WNT3A/RSPO2 (W3AR2) co-treatment also enhanced colony formation, further suggesting that WNT/RSPO cooperation underlies increased colony formation (Appendix A).

We next examined the colony-forming ability of DLESPs lacking functional *Rspo2*. DLESPs were isolated from *Rspo2^flox/flox^*;*CAG-Cre^ESR^* mice, in which *Rspo2* function was conditionally inactivated by ex vivo 4-OHT treatment (*Rspo2^CKO^*) (Appendix A). We confirmed that 4-OHT treatment for the first two days of culture effectively disrupted *Rspo2* (Appendix A). *Rspo^CKO^* DLESPs exhibited poor colony-forming ability after 4-OHT treatment, whereas untreated *Rspo2^flox/flox^* DLESPs formed colonies effectively (Figure 2F,G). We ruled out the possibility that 4-OHT treatment itself produced any inhibitory effect on colony formation, because control DLESPs isolated from *Rspo2^flox/flox^* mice lacking the *CAG-Cre^ESR^* transgene showed a similar efficiency of colony formation in both untreated and 4-OHT-treated cultures (Figure 2F,I). When *Rspo2^flox/flox^*;*CAG-Cre^ESR^* DLESP organoids were treated with 4-OHT on days 6 and 7 of culture, loss of *Rspo2* still caused a deficit in the colony-forming ability of DLESP organoids (Figure 2F,H). We further determined whether the recombinant RSPO2 protein could rescue the colony formation defect in *Rspo2^CKO^* organoids. RSPO2 or RSPO2/WNT3A recombinant proteins were able to partly restore the colony-forming ability of *Rspo2^CKO^* DLESPs (Figure 2J,K). Taken together, our results suggest that RSPO2 is a key autocrine factor that regulates the expansion of DLESPs ex vivo.

### 2.3. RSPO2 Supports Colony Formation of Club Cells through the Activation of Endogenous WNT/β-Catenin Signaling

To further define the specific role of RSPO2 in club cells, we utilized ex vivo organoid culture using EpCAM^+ve^;SSEA1^+ve^ club cells. Club cells of approximately 90% purity were isolated. RSPO2 when administered to the organoid culture starting from the first day of culture and two weeks after starting the culture significantly increased the colony-forming efficiency of club cells in both cases (Figure 3A,B). DLESP organoid formation depends on the presence of fibroblasts, and presumably fibroblast-derived factors, in culture. Without fibroblasts, DLESPs could not form colonies even in the presence of WNT3A, RSPO2, or W3AR2 (Appendix A). When we included FGF10 and HGF, which can replace fibroblasts in the culture media, RSPO2 significantly enhanced the colony-forming ability of club cells under fibroblast-free conditions (Figure 3C,D).

In contrast, club cells lacking functional *Rspo2* displayed impaired colony-forming ability (Figure 3E,F). We found a similar colony-forming defect in *Rspo2^CKO^* club cells in a fibroblast-free, FGF10/HGF supplemented organoid culture (Appendix A). However, on ablating *Rspo2* on day 4 of organoid culture, colony-forming defects of *Rspo2^CKO^* club cells largely disappeared (Figure 3E,F), suggesting that endogenous *Rspo2* function for club cell colony formation is critical at the early stage of organoid culture and may not be essential in the mid-to-late stage of organoid culture. Nonetheless, our results convincingly suggest that RSPO2 plays a positive role in the expansion of club cells in ex vivo 3D organoid cultures.

We further determined how RSPO2 regulated colony formation in club cells. RSPO2 treatment apparently activated canonical WNT signaling, as the expression of WNT signaling downstream target genes, *Axin2* and *Lgr5*, was elevated (Figure 3G). Moreover, RSPO2 treatment profoundly increased the expression of active β-catenin in club cell organoids (Figure 3H). Since RSPO2 was recently demonstrated to inhibit BMP signaling [44], we examined the expression of BMP downstream target genes (Appendix A). There were no significant changes in *Bmpr1a*, *Id1*, and *Smad6* expression by RSPO2, ruling out the possibility of BMP signaling inhibition by RSPO2 in club cell organoids.

When endogenous secretion of WNT ligands from either club cells or co-cultured fibroblasts was inhibited by the porcupine inhibitor IWP-2, colony formation enhanced by RSPO2 was reversed (Figure 3I,J). These results further support the notion that RSPO2 produces its activity via the activation of canonical WNT signaling. Collectively, these results suggest that club cell-derived RSPO2 in combination with club cell-derived WNTs regulates colony formation and cell expansion via activation of canonical WNT signaling but not inhibition of BMP signaling.

### 2.4. RSPO2 Promotes the Differentiation of Club Cells into Multiple Epithelial Lineages

In DLESP organoids, we found that RSPO2 treatment increased the expression of the AT2 cell marker, *Sftpc*, while decreasing the expression of the club cell marker, *Scgb1a1* (Appendix A). To determine how RSPO2 affects club cell differentiation in the ex vivo culture, we analyzed the RNA expression of several different lineage markers in club cell organoid RNA samples by qRT-PCR. We found that RSPO2 treatment did not significantly change the expression of club cell markers, *Scgb1a1*, *Cyp2f2*, and *Bpifa1* (Figure 4A). In contrast, RSPO2 considerably increased the expression of the ciliated cell marker *Foxj1* and the goblet cell marker *Muc5ac* with no significant expression change of the basal cell marker *Krt5* (Figure 4B). Interestingly, RSPO2 treatment significantly increased the expression of AT2 cell markers, *Sftpc* and *Abca3*, and the AT1 cell marker *Rtkn2*, while it decreased the expression of mature AT1 markers *Aqp5* (Figure 4C,D).

In *Rspo2^CKO^* organoids, the expression of the club cell marker *Scgb1a1* was increased (Figure 4E). On the other hand, the expression of both ciliated cell and basal cell markers, *Foxj1* and *Krt5*, was decreased in *Rspo2^CKO^* organoids (Figure 4E). *Rspo2* knockdown also increased the expression of *Sftpc*, an AT2 cell marker, while no change was observed in the AT1 cell marker, *Rtkn2* expression (Figure 4E). Moreover, using immunofluorescence staining for SCGB1A1, FOXJ1, and SFTPC, RSPO2 treatment decreased the SCGB1A1^+ve^ cell number and increased the FOXJ1^+ve^ and SFTPC^+ve^ cell number per organoid (Figure 4F,G). These results suggest that *Rspo2* is involved in the differentiation of club cells into multiple epithelial lineages.

### 2.5. RSPO2 Promotes the Differentiation of Club Cells into Alveolar Lineages in an Endogenous WNT/β-Catenin-Activation Manner

To determine the role of canonical WNT signaling potentiated by RSPO2 on club cell differentiation, we blocked endogenous WNT secretion using IWP-2, which is a condition that reverses RSPO2 effects, thereby blocking canonical WNT signaling potentiation by RSPO2. Using qRT-PCR analysis, we confirmed that the inhibition of RSPO2-potentiated canonical WNT signaling after IWP-2 treatment, as indicated by the severely decreased expression of WNT downstream targets, *Axin2* and *Lgr5* (Figure 5A,B). RSPO2 treatment alone decreased the expression of club cell markers *Scgb1a1* and *Cyp2f2,* while RSPO2/IWP-2 co-treatment reversed the expression of both *Scgb1a1* and *Cyp2f2* (Figure 5C,D). Interestingly, the elevated expression of ciliated cell markers *Foxj1* and *Tubb4a* was not reduced by IWP-2 co-treatment (Figure 5E,F). These results indicate that RSPO2 may play a negative regulatory role in maintaining club cell marker expression via canonical WNT signaling while inducing ciliated cell differentiation in a canonical WNT-independent manner. Moreover, IWP-2 treatment significantly reversed the expression of AT2 cell markers, *Sftpc* and *Sftpa,* and the AT1 marker, *Rtkn2*, induced by RSPO2 (Figure 5G–I), suggesting that RSPO2 promotes the differentiation of club cells into AT2 cells through the activation of canonical WNT signaling.

### 2.6. Rspo2 Is Essential for the Colony Formation and Differentiation of AT2 Cells

The RSPO2-driven alveolar lineage differentiation of DLESPs and club cells and its expression in AT2 cells led us to investigate whether there is a specific role of RSPO2 in AT2 cell function. We isolated AT2 cells with over 90% purity (Appendix A). RSPO2 treatment slightly, but significantly, increased the colony-forming efficiency of AT2 cells in ex vivo 3D organoid cultures (Figure 6A,B). Moreover, RSPO2 considerably enhanced the colony-forming efficiency of AT2 cells in a fibroblast-free, FGF10/HGF-supplemented organoid culture (Figure 6C,D). When *Rspo2* was ablated on the first day of culture, AT2 cells displayed impaired colony-forming ability (Figure 6E,F). In contrast, when *Rspo2* inactivation was performed on day 5 of organoid culture, the *Rspo2* knockout AT2 cells showed substantial colony-forming ability. These results suggest that *Rspo2* appears to be important for AT2 cell colony formation, mainly in the early stages of culture.

To assess the differentiation of AT2 cells by RSPO2, we performed qRT-PCR analysis for different alveolar epithelial markers. RSPO2 treatment significantly increased the expression of AT2 cell markers, *Sftpc* and *Sftpa,* and the AT1 marker, *Rtkn2,* while decreasing the expression of the AT1 marker, *Aqp5* (Figure 6G). This was consistent with AT1/AT2 marker expression in DLESP- and club cell-derived organoids. Moreover, cells positive for the expression of HOPX, an early AT1 marker, were also significantly increased by RSPO2 in AT2-derived organoids (Figure 6H,I). In support of the RSPO2 treatment results, the expression of AT2 markers, *Sftpc* and *Sftpa*, and AT1 markers, *Rtkn2* and *Aqp5*, declined in *Rspo2^CKO^* organoids (Figure 6J). These results convincingly suggest that RSPO2 actively regulates both the self-renewal and differentiation of AT2 cells in a cell-autonomous manner.

### 2.7. Loss of the Rspo2 gene Impairs Airway Regeneration after Naphthalene-Induced Lung Injury

Specific *Rspo2* expression in club cells led us to investigate the function of *Rspo2* in airway regeneration. We utilized a well-established naphthalene-induced lung injury model in which most club cells are depleted by 2 days after injury (dai) and restored by 21 dai [4,45,46]. Using hematoxylin and eosin staining, immunofluorescence staining, and qRT-PCR analysis, we observed the depletion and restoration of club cells in both the proximal and terminal airways (Appendix A). In addition, ciliated cells also disappeared in the proximal airways, confirming effective injury and regeneration (Appendix A).

*Rspo2* RNA expression significantly decreased during the first week after injury but was fully restored by 14 dai (Figure 7A). We detected faithful RSPO2 protein expression in the surviving and regenerating club cells during the regeneration process using immunofluorescence staining (Figure 7B). Interestingly, RSPO2 protein expression preceded SCGB1A1 expression at 4–7 dai, suggesting that RSPO2 secreted from surviving club cells may be a key autocrine and paracrine factor to enhance club cell expansion and regeneration processes.

To determine the function of *Rspo2* during airway regeneration, we injured the lungs of *Rspo2^CKO^* and control wild-type mice by naphthalene administration. The efficiency of *Rspo2* knockout was confirmed by genomic DNA PCR using specific primer sets and immunofluorescence staining (Appendix A). Interestingly, *Rspo2* ablation was more effective in alveolar epithelial cells than in the airway epithelium (Appendix A). At day 14 after injury, we found that the thickness of airway bronchial epithelium was significantly decreased in *Rspo2^CKO^* mice compared to that in the control wild-type mice (Figure 7C,D). Furthermore, the number of SCGB1A1^+ve^ club cells was significantly decreased in *Rspo2^CKO^* mice (Figure 7C,E). Interestingly, no significant difference was detected in the number of Ac-α-tubulin^+ve^ ciliated cells in both *Rspo2^CKO^* and wild-type mice (Figure 7C,F). Moreover, we found a decrease in the number of Ki67^+ve^ cells in the bronchial epithelium of *Rspo2^CKO^* mice (Figure 7G,H). We also found that active β-catenin was reduced in the bronchial epithelium of *Rspo2^CKO^* mice, suggesting a reduction in WNT/β-catenin signaling after *Rspo2* ablation. These results suggest that *Rspo2* is important for club cell regeneration and may play a role in the activation of the WNT/β-catenin signaling pathway.

## 3. Discussion

### 3.1. Rspo2 Expression in the Adult Lung Stem/Progenitor Cells

The importance of the RSPO protein in the regulation of WNT signaling, especially canonical WNT/β-catenin signaling, is immense [31]. Therefore, the expression and function of RSPOs should be determined under any biological condition where canonical WNT signaling is expected to play a critical role. In the adult lungs, WNT signaling is important for homeostasis and regeneration [1,22,23]. Our understanding of RSPO function in conjunction with canonical WNT signaling in the adult lung, especially stem/progenitor cells, is very limited.

Through varied analyses of *Rspo2* RNA and RSPO2 protein expression, we convincingly demonstrated that *Rspo2* is specifically expressed in the lung epithelial stem/progenitor cells, including club cells within the airways and AT2 cells in the alveoli of the mouse adult lungs. Interestingly, we failed to detect any meaningful *Rspo2* expression in lung mesenchymal cells, which are a compartment for *Rspo2* expression in the embryonic lungs [38,39].

Since *Rspo2* RNA expression in the embryonic lungs was examined by a whole-mount in situ hybridization technique [38], it is not clear whether any specific epithelial cell population expresses *Rspo2* during embryonic lung development. However, mesenchymal cell-specific *Rspo2* expression is distinctly extinguished in the adult lungs. How and when *Rspo2* expression shifts from mesenchymal to epithelial cells in embryonic and adult lungs are fundamentally intriguing questions to better understand lung development in mice.

Club cells and AT2 cells have been previously shown to be associated with canonical WNT/β-Catenin signaling activity [1,3]. Several WNT ligands, including WNT3A, WNT5A, and WNT7B, are upregulated in club cells after naphthalene-induced airway injury [17]. It has also been demonstrated that canonical WNT signaling is activated in AT2 cells after bleomycin-induced injury in mice [47,48]. The recently identified *Axin2^+ve^* AT2 subpopulation showing stem/progenitor activity represents canonical WNT signaling recipients within alveoli [26,27,49]. Pericytes, endothelial cells, and LGR5^+ve^ stromal cells within alveolar compartments express WNT3A and WNT5A ligands [17,50,51]. Therefore, RSPO2 expression in these cell populations strongly indicates that RSPO2 is a key autocrine/paracrine factor that positively regulates canonical WNT signaling in cooperation with WNT ligands derived from the same cell population and niche cells.

### 3.2. Autocrine RSPO2 Supports Colony Formation of Lung Stem/Progenitor Cells

Consistent with this expectation, our experimental data collectively support the conclusion that RSPO2 is an autocrine factor that specifically regulates the expansion of lung epithelial stem/progenitor cells. First, RSPO2 treatment effectively enhanced the colony-forming efficiency of naive DLESPs, a cell population containing AT2 cells, club cells, and other epithelial cells, in organoid cultures. In a similar organoid culture utilizing highly purified club cells or AT2 cells, RSPO2 showed pro-proliferative activity similar to that of the tested cells. Although RSPO2 alone cannot support the colony formation of these cells in a fibroblast feeder-free organoid culture, it was sufficient to enhance colony formation when FGF10 and HGF, two essential factors required for DLESP-, club cell-, or AT2 cell-derived organoid culture, were provided.

The second evidence for RSPO2 as an autocrine factor was obtained from *Rspo2* knockout organoids and mice. The loss of endogenous *Rspo2* resulted in reduced or poor colony-forming ability of primary DLESPs, club cells, and AT2 cells. The reduced colony-forming efficiency in *Rspo2^KO^* DLESPs was partially restored by RSPO2 supplementation. Consistent with these ex vivo data, mice carrying conditionally inactivated *Rspo2* showed severe delay in the regeneration of the epithelial cell layer in the airways after naphthalene-induced airway injury, further supporting the essential role of RSPO2 in the proliferation of lung epithelial stem/progenitor cells.

The pro-proliferative activity of RSPO2 on DLESPs, club cells, or AT2 cells is mainly mediated by the activation of canonical WNT signaling and is largely dependent on the endogenous WNT ligands produced by these cells or lung fibroblast feeders in organoid culture. RSPO2 treatment induced the expression of canonical WNT signaling target genes, such as *Axin2* and *Lgr5*, and increased the number of cells expressing an active form of β-Catenin in these organoids. Co-treatment with RSPO2 and WNT3A enhanced the colony-forming ability of DLESPs in the organoid culture in a synergistic manner, which is a typical outcome resulting from the cooperative activation of canonical WNT signaling induced by RSPO and WNT. Furthermore, the addition of the porcupine inhibitor, IWP-2, in the culture, which can block the secretion of endogenous WNT ligands, significantly reversed the RSPO2 effect.

Previously, the club cell-specific deletion of *Ctnnb1* that encodes β-Catenin did not affect the proliferation or differentiation of club cells following naphthalene-induced injury [52]. Thus, β-Catenin appears to be dispensable for club cell function during the regeneration of the bronchiolar epithelium. Recently, it was reported that club cells are enriched within a subpopulation of DLESPs characterized by low WNT/β-Catenin activity (called WNT^low^ cells) [53], further supporting the dispensable role of β-Catenin in club cells. However, our data seem to support the active role of canonical WNT signaling in cell proliferation in club cell organoids. Induction of the active form of β-Catenin by RSPO2 in club cell organoids suggests that club cells are responsive to the activation of canonical WNT signaling. It is possible that canonical WNT signaling activated by RSPO2 may transmit the signal in a β-Catenin-dependent and independent manner. Moreover, β-Catenin-dependent activation may have a lesser role and may be compensated by β-Catenin-independent activation. Therefore, future investigations are warranted.

Unlike club cells, our data utilizing AT2 cell organoids are consistent with the active role of canonical WNT signaling that has been established by a number of earlier studies. Canonical WNT signaling within AT2 cells was reported to be relatively low in uninjured normal lungs, but it was significantly activated after lung injury [47,48]. In ex vivo organoid cultures of AT2 cells with alveolar stromal cells, WNT3A treatment enhanced organoid formation [17] similar to that shown in our data from AT2 cell organoid culture. Interestingly, AT2 cells consist of heterogeneous populations in terms of WNT responsiveness. *Axin2^+ve^* AT2 (WNT-responsive) cells, a distinct AT2 sub-population, represent evolutionarily conserved alveolar epithelial stem/progenitor cells [26,27]. Consistent with their WNT responsiveness, active canonical WNT signaling is important for the expansion of *Axin2^+ve^* AT2 cells during alveolus regeneration [26,27,49]. Severe alveolar epithelial cell death and deficits in alveolar regeneration were detected upon bleomycin-induced injury in epithelium-specific *Ctnnb1* knockout mouse lungs [47,48]; thus, β-Catenin function appears to be indispensable in AT2 cells.

### 3.3. Diverse Regulation of Differentiation by RSPO2 in the Lung Epithelial Progenitor/Stem Cells

In addition to its pro-proliferative activity, RSPO2 differentially influences the differentiation process in lung stem/progenitor cells. RSPO2 decreased the number of club cells positive for SCGB1A1 in DLESP organoids. This was also true for club cell-derived organoids. In contrast, both the expression of ciliated cell markers, *Foxj1,* and the number of FOXJ1 positive cells was increased by RSPO2 in club cell-derived organoids. Consistent with these results, an opposite *Foxj1* gene expression pattern was observed in club cell organoids lacking *Rspo2*.

Unexpectedly, RSPO2 treatment in the presence of IWP-2 (no endogenous WNT condition) could recover the decreased expression of club cell markers caused by either RSPO2 or IWP-2 treatment alone to a level equivalent to or higher than the control level. In addition, while RSPO2 treatment increased the expression of ciliated cell markers, IWP-2 also increased the expression of the same markers in club cell organoids. These results raise the possibility that RSPO2 and WNT may play differential and independent roles in the maintenance and differentiation of club cells. In the absence of WNT, RSPO2 may function through another signaling pathway to promote club cell self-renewal. The RSPO2-mediated increase in ciliated cell differentiation may also be mediated by a WNT-independent pathway. Further investigation is needed to determine whether the WNT^low^ club cell population is the cell type associated with the WNT-independent RSPO2 effect.

RSPO2 increased the expression of the AT2 cell marker *Sftpc* in the DLESP organoid. AT2-and AT1-specific marker expression was elevated, whereas mature AT1 marker expression was reduced. Since DLESPs contain AT2 cells and RSPO2 enhances the colony-forming ability of purified AT2 cells in organoid culture, AT2 marker expression by RSPO2 in DLESPs is likely a consequence of AT2 cell expansion. Similarly, both AT1 and AT2 cell marker expression are differentially regulated by RSPO2 in club cell organoids. Club cells have the ability to generate alveolar organoids containing AT1 and AT2 in vitro [12] as well as contribute to alveolar repair after bleomycin-induced injury by generating AT1 and AT2 cells [10]. Interestingly, the activation of WNT/β-Catenin signaling in WNT^low^ DLESPs significantly induced alveolar-type organoids [53]. Therefore, the WNT^low^ population of DLESPs may contribute to the AT2 marker expression induced by RSPO2.

In AT2 cell organoids, RSPO2 induced the expression of AT2 and intermediate AT1 markers while inhibiting mature AT1 cell marker expression. AT2 cells lacking *Rspo2* showed severely reduced expression of both AT2 and intermediate AT1 cell markers. Consistent with our study, RSPO1/2 protein treatment reportedly induced the expression of alveolar epithelial markers in murine lung epithelial cells [54]. Furthermore, previous studies demonstrated that active WNT signaling in AT2 cells, especially AXIN2^+ve^ AT2 cells, inhibited AT2 to AT1 cell differentiation [26,27,49].

### 3.4. Is RSPO2 a Paracrine Factor for Other Types of Lung Cells?

We did not investigate this possibility in the present study. Our preliminary cell population analysis in naphthalene-injured lungs of *Rspo2* knockout mice clearly indicates that loss of *Rspo2* caused significant changes in many cell types other than epithelial cells, supporting the paracrine role of RSPO2. In contrast to this assumption, it has been reported that RSPO1 and 2 treatment induces β-Catenin-dependent gene transcription in murine lung epithelial cells but not in murine lung fibroblasts, as assessed by a canonical WNT reporter, TOPflash, luciferase assay [54]. Interestingly, it was recently reported that RSPO2 protein expression is detected in bronchiolar and alveolar epithelial cells in human idiopathic pulmonary fibrosis lungs, and RSPO2 has anti-proliferative and apoptotic effects on lung fibroblasts [40]. RSPO2 also inhibited collagen production and increased the expression of matrix metalloproteinase (MMP)-1. *RSPO2* knockdown using short hairpin RNA in these cells induced the opposite effects. Thus, the effects of RSPO2 on lung fibroblasts remain unclear.

Recently, LGR5^+ve^ and LGR6^+ve^ mesenchymal cells have been identified in ASMCs surrounding the airway epithelia and alveoli, respectively [17]. Since LGR5 and LGR6 are the cognate receptors for RSPO2, these cells are potential targets for RSPO2 derived from club and AT2 cells. LGR6^+ve^ ASMCs are very important during airway regeneration, promoting airway differentiation of epithelial progenitors through WNT-FGF10 cooperation, and the depletion of these cells caused major defects in regeneration following naphthalene-induced lung injury in vivo. LGR5^+ve^ mesenchymal cells are WNT-producing cells in the alveoli, particularly WNT3A and WNT5A, and support the alveolar differentiation of epithelial progenitors through WNT signaling activation. Whether club cells or AT2 cell-derived RSPO2 are directly involved in the regulation and function of LGR5^+ve^ and LGR6^+ve^ mesenchymal cells is an intriguing and important question to be explored in the future.

## 4. Materials and Methods

### 4.1. Animals

Eight to twelve-week-old wild-type ICR and C57BL/6N mice were used for the isolation of primary lung stem/progenitor cells and fibroblasts and in vivo regeneration experiments, respectively. *Rspo2^LacZ/+^* mice [43] maintained in our laboratory were used for β-galactosidase staining and immunohistochemistry. Mice carrying a conditional *Rspo2* knockout allele (*Rspo2^flox^*) were generated in our laboratory (see below). *Tg*(*CAG-cre/Esr1**)*^5Amc^*(hereafter, *CAG-Cre^ESR^*) mice in which transgenic *Cre* expression in the whole body can be conditionally achieved by tamoxifen administration were obtained from the Jackson Laboratory (MGI:2182767, Bar Harbor, ME, USA). Compound *Rspo2^flox/flox^*; *CAG-Cre^ESR^* mice were produced by mating *Rspo2^flox/flox^* and *Rspo2^flox/+^*; *CAG-Cre^ESR^* mice. Mice were housed in a specific pathogen-free animal facility, and animal handling and experimental procedures were approved by the Institutional Animal Care and Use Committees of the Maine Medical Center Research Institute and Soonchunhyang University.

### 4.2. Gene Targeting for Conditional Rspo2 Inactivation

A conditional *Rspo2* knockout vector was constructed in the pGKneoF2L2DTA targeting vector, kindly provided by P. Soriano (Appendix A). Both short (3.5 kb) and long (6.5 kb) arm DNA fragments of the *Rspo2* gene were amplified by PCR using C57BL/6J genomic DNA. Two loxP sites were inserted into the 5′ and 3′ flanking regions of exon 2 encoding the start codon of *Rspo2*, whereby Cre recombination would result in a complete inhibition of RSPO2 protein translation. The *Rspo2^flox^* targeting vector plasmid DNA was transfected into the mouse embryonic stem cell line derived from an F1 hybrid (C57BL/6J × 129sv) mouse at the Center for Mouse Genome Modification at the University of Connecticut Health. Neomycin-resistant ES clones were screened by genomic PCR, and the properly targeted ES clones were injected into Balb/c blastocytes to produce chimeric mice in the Mouse Transgenic and Genome Modification facility at the Maine Medical Center Research Institute. Chimeric mice were mated to C57BL/6J mice, and genomic DNA of their progeny was genotyped by PCR to confirm the germline transmission of the targeted gene *Rspo2*.

### 4.3. In Vivo Naphthalene-Induced Lung Injury

Eight-week-old wild-type C57BL/6N or *Rspo2^flox/flox^*;*CAG-Cre^ESR^* male mice were injected intraperitoneally with naphthalene (275 mg/kg body weight). To conditionally ablate *Rspo2* before injury, 75 mg/kg of tamoxifen (TMX; 20 mg/mL in corn oil; Sigma-Aldrich, St. Louis, MO, USA) or corn oil were injected intraperitoneally into *Rspo2^flox/flox^*;*CAG-Cre^ESR^* mice every 24 h for five consecutive days starting from day 10 before naphthalene injury. The lungs were harvested at different time points after injury, as indicated.

### 4.4. Isolation of Primary Lung Stem/Progenitor Cells and Fibroblasts

After opening the thoracic cavity of mice, the lungs were perfused with 10 mL of sterile cold phosphate-buffered saline (PBS) through the right ventricle of the heart. The exposed trachea was cannulated with a 22G catheter and injected with 2 mL of digestion solution (PBS containing 80 U/mL collagenase type II (Worthington Biochemical, Lakewood, NJ, USA) and 2 U/mL Dispase II (Sigma-Aldrich)), which was followed by clogging with forceps for 2 min. The lung lobes were dissected from the trachea, finely minced, and incubated with 20 mL digestion solution for 1 h in a 37 °C incubator with shaking followed by passaging through an 18G needle 4–5 times and an additional 15 min incubation at 37 °C after adding 70–100 units of DNAase I (Sigma-Aldrich). After passing through a 21G needle, the cell suspension was filtered through 100 and 40 μm mesh cell strainers and centrifuged at 400× g for 5 min to collect the isolated cells. Red blood cells were lysed using lysis buffer (Sigma-Aldrich). Hematopoietic (CD45^+ve^) and endothelial (CD31^+ve^) cells were depleted using magnetic cell separation (MACS). From CD45^−ve^;CD31^−ve^ cells, CD326 (EpCAM)^+ve^ distal lung epithelial cells (DLESPs) and EpCAM^+ve^;SSEA1^+ve^ club cells [17,55], were isolated using FACS Aria III (BD Biosciences, San Jose, CA, USA) at the Soonchunhyang Biomedical Science Core Facility Center of Korea Basic Science Institute. The purity of the isolated DLESP and club cells was confirmed by cell-specific marker expression using immunofluorescence staining.

AT2 cells were isolated as previously described [56] with some modifications. Briefly, 10 mL of sterile cold PBS was injected into the right ventricle of the heart to remove blood from the lungs. Two milliliters of Dispase II (2 U/mL in PBS) and 600 μL of 1% low-melt agarose (45 °C) were injected into the lungs through the 22G catheter cannulated into the trachea. Then, the lungs were covered with crushed ice for 2 min to allow the injected agarose to solidify. Dissected lung lobs were incubated with 20 mL Dispase II (2 U/mL in PBS) for 45 min at 29 °C in an incubator with rocking. The lung lobes were transferred into 100 mm Petri dishes containing 20 mL of medium (DMEM, 10% FBS, 1× penicillin/streptomycin, 10 mM HEPES, and 70 U DNAase I). The lung parenchyma was separated from the large airways and incubated for another 10 min at 29 °C with rocking to complete enzymatic digestion. Cells were sequentially filtered through 100 and 40 μm mesh cell strainers, and after the lysis of red blood cells, hematopoietic and endothelial cells were depleted by MACS as described above. AT2 cells were sorted as EpCAM^+ve^ cells using FACS Aria III. The purity of the isolated AT2 cells was confirmed by immunofluorescence staining of the AT2 cell marker, SFTPC.

Lung fibroblasts were isolated from CD45^−ve^;CD31^−ve^ cell suspension. The cell suspension was plated on uncoated plastic dishes in fibroblast culture media (DMEM 10% FBS, 1X penicillin/streptomycin, and 2.5 μg/mL amphotericin B). Approximately 24 h later, the culture medium was replaced to remove the unattached cells. Lung fibroblasts were either passaged for immediate use or frozen in 10% DMSO media and stored in a liquid nitrogen tank until use.

### 4.5. Ex Vivo 3D Organoid Culture

A total of 2–5 × 10^3^ DLESPs, club cells, or AT2 cells mixed with primary lung fibroblasts in a 1:5 ratio were resuspended in CFU-Epi media (DMEM/F12, 10% FBS, 1× penicillin/streptomycin, 2.5 µg/mL amphotericin B, 1× Insulin-Transferrin-Selenium-X (Gibco, Grand Island, NY, USA), and 2 µg/mL heparin sodium salt) [57]. Cell suspension and growth factor-reduced phenol red-free Matrigel (Corning, Oneonta, NY, USA) were mixed in a 1:1 ratio. Routinely, 90 μL of cell mixture was placed on ThinCert^TM^ 0.4 μm pore size transparent PET membrane transwell inserts (Greiner Bio-One, Frickenhausen, Germany) in 24-well plates, and 600 μL of CFU-Epi medium was placed in the lower chamber and cultured up to 14 days. The medium in the lower chamber was changed daily unless stated otherwise. In fibroblast-free culture, 2–4 × 10^4^ DLESPs, club cells, or AT2 cells were resuspended in CFU-Epi medium supplemented with 50 ng/mL FGF10 (Peprotech, Rocky Hill, NJ, USA) and 30 ng/mL hepatocyte growth factor (Peprotech), and the organoids were cultured as described above. Recombinant R-spondin2 (200 ng/mL; Peprotech) and WNT3A (20 ng/mL; Peprotech) proteins, IWP-2 (5 μM; Sigma-Aldrich), and 4-hydroxytamoxifen (4-OHT; 500 nM; Sigma-Aldrich) were directly added to the medium in the lower chamber for the indicated period. The number of colonies per insert was counted on days 7 and 14 of the organoid culture or as indicated.

### 4.6. Sample Preparation for Histological and Immunological Staining

Matrigel 3D organoids were processed for histological analysis as previously described [58] with some modifications. Briefly, Matrigel 3D colonies were fixed in 4% paraformaldehyde (PFA) overnight at 4 °C. Τhe membrane of the transwell containing the Matrigel organoids was cut and embedded into the liquefied HistoGel (Thermo Fisher Scientific, Waltham, MA, USA). The HistoGel molds were dehydrated in an ascending series of ethanol (70–100%, Carlo Erba Reagents, Peypin, France) and embedded in paraffin.

The lungs were perfused with 10 mL of sterile cold PBS to remove blood and then inflated with 2 mL of 10% neutral-buffered formalin (NBF). The trachea was clogged with a thread of cloth, and the lungs were fixed in 10% NBF for 48 h at 24 °C. The fixed lungs were dehydrated in an ascending series of ethanol (70–100%) and embedded in paraffin. Then, 6 μm thick paraffin-embedded lung or organoid sections were deparaffinized and rehydrated by placing them in a series of xylene and ethanol washes for in situ RNA and immunofluorescence staining.

Freshly isolated EpCAM^+ve^, club, or AT2 cells were centrifuged at 400× *g* for 5 min at 4 °C, which was followed by aspiration of the supernatant. Cell pellets were resuspended in 0.1 mL Hank’s balance salt solution (Thermo Fisher Scientific) containing 0.2% bovine serum albumin. Cells were attached to slides (Muto Pure Chemicals, Tokyo, Japan) using Cytospin 4 (Thermo Fisher Scientific) at 110× *g* for 5 min. The slides were air-dried and fixed with 4% PFA for 15 min at 24 °C.

### 4.7. Immunofluorescence Staining, In Situ RNA Hybridization, and Histological Staining

For immunofluorescence staining, antigen retrieval was performed by boiling the slides in citrate buffer (10 mM citric acid, 0.05% Tween-20, pH 6.0) for 20 min, and then, the slides were allowed to cool at 24 °C for at least 30 min. The slides were washed with PBS and permeabilized with 0.2% Triton-X for 15 min. Following blocking in 5% normal goat or donkey serum in PBS containing 0.2% Triton-X for 2 h at 24 °C, the slides were incubated with the primary and secondary antibodies listed in Appendix A. Stained slides were mounted with fluoroshied mounting medium with DAPI (Sigma-Aldrich). In situ RNA hybridization was performed using the RNAscope 2.5 Assay kit (Advanced Cell Diagnostics, Newark, CA, USA) on formalin-fixed paraffin sections [59] following the manufacturer’s instructions.

Whole-mount X-gal staining was performed as previously described [60]. Briefly, after removing the blood as described previously, the lung lobes were fixed in 2% formaldehyde in PBS at 24 °C for 30–60 min. The lung lobes were washed with PBS three times and then incubated in pre-warmed, freshly prepared X-gal staining solution (1 mg/mL 5-bromo-4-chloro-3-indolyl-b-D-galactoside, 5 mM K_3_Fe(CN)_6_, 5 mM K_4_Fe(CN)_6_, 2 mM MgCl_2_, 0.02% NP-40, 0.01% Na-deoxycholate in PBS, pH 7.4) overnight in the dark at 37 °C. Then, the lobes were postfixed in 4% PFA, dehydrated in an ascending series of ethanol (70–100%), and embedded in paraffin. Paraffin-embedded sections were deparaffinized by placing them in a series of xylene and ethanol washes and then rehydrated. The stained slides were imaged using a DMi8 inverted microscope (Leica Microsystems, Germany).

For hematoxylin and eosin staining, slides were stained with Gill’s II Hematoxylin solution (Sigma-Aldrich) for 1 min and then washed under running tap water. After bluing in 0.2% ammonium hydroxide, they were stained with eosin Y solution (Sigma-Aldrich). After dehydration in 70%, 95%, and 100% ethanol series and clearance with HistoChoice^®^ clearing agent (Sigma-Aldrich), the stained slides were mounted in VectaMount permanent mounting media (Vector Laboratories, CA, USA).

Cell images were obtained using a Ti-U inverted microscope (Nikon Instruments Inc, Japan), DMi8 inverted microscope (Leica Microsystems), or LSM 710 confocal microscope (Zeiss, Germany) at the Soonchunhyang Biomedical Science Core Facility Center of Korea Basic Science Institute.

### 4.8. Western Blot Analysis

Cells were lysed with freshly prepared 1× RIPA lysis buffer (20 mM Tris (pH 7.5), 150 mM NaCl, 1% TritonX-100, 0.1% sodium deoxycholate) containing 1 mM PMSF, 100 mM NaF, 1 mM Na_3_VO_4_, and 1X protein inhibitor cocktail (Calbiochem, San Diego, CA, USA). Protein samples were quantified using the SMART™ BCA Protein Assay Kit (iNtRON, Seongnam-si, Korea). Equal amounts of protein were loaded, and Western blot analysis was performed as described previously [61]. The following primary antibodies were used: RSPO2 (rabbit, 1:1000; Proteintech), SFTPC (rabbit, 1:1000; EMD Millipore, Burlington, MA, USA), and β-actin (mouse, 1:5000; Sigma-Aldrich). The following secondary antibodies were used: HRP-goat anti-mouse IgG (1:5000; AB Frontier, Seoul, Republic of Korea) and HRP-goat anti-rabbit IgG (1:10,000; Sigma).

### 4.9. Quantitative PCR (qRT-PCR) Analysis

Total RNA was isolated from lung tissue, freshly isolated lung primary cells, or lung 3D organoids using TRIzol reagent (Thermo Fisher Scientific). First-strand cDNA was synthesized from 0.4 to 1 μg total RNA using a ProtoScript first-strand cDNA synthesis kit (New England BioLabs, Ipswich, MA, USA). cDNA equivalent to 10 or 20 ng of total RNA was used for each qRT-PCR reaction. The PCR primer sequences used in this study are listed in Appendix A.

### 4.10. Statistical Analysis

All experiments were performed at least in triplicate, unless otherwise stated. The results are presented as mean ± SEM (standard error of the mean). The experimental groups were compared using a non-paired Student’s t-test using GraphPad Prism software (GraphPad Software, La Jolla, CA, USA). Statistical significance was set at *p* < 0.05.

## 5. Conclusions

Our study convincingly demonstrated the critical autocrine role of RSPO2 as a canonical WNT signaling regulator in DLESPs, including club cells and AT2 cells. RSPO2 expressed in club cells and AT2 cells supports their expansion and differentially regulates the differentiation of these cells mainly through the activation of canonical WNT signaling. Our findings suggest that RSPO2 is a key regulator of lung homeostasis and regeneration and may play a protective role in acute lung tissue damage and chronic lung diseases.

## Figures and Tables

**Figure 1 ijms-23-03089-f001:**
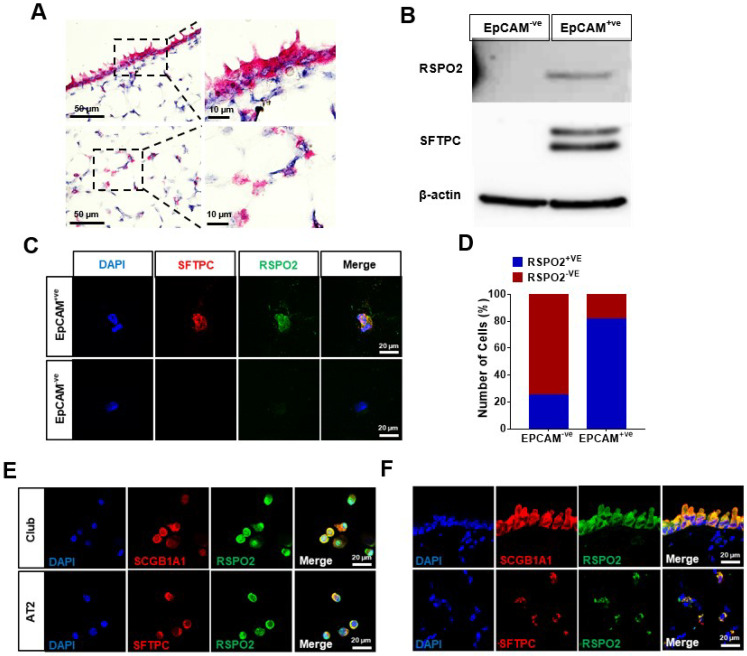
RSPO2 is expressed in club and AT2 cells of adult lungs. (**A**) Single-molecule RNA in situ hybridization detection of *Rspo2* RNA. Top and bottom panels represent the airway and alveoli, respectively. The right panels show the magnified images of the dotted rectangular region of the left panels. (**B**) Western blot analysis of RSPO2 protein expression in freshly isolated EpCAM^+ve^ distal lung epithelial cells. SFTPC and β-actin expression were also determined for validating cell purity and as protein loading control, respectively. (**C**) Immunofluorescence staining of RSPO2 (green) and AT2 cell marker, SFTPC (red), in freshly isolated EpCAM^+ve^ distal lung epithelial cells. (**D**) Quantification of RSPO2^+ve^ cell fraction in the isolated distal lung epithelial cells. (**E**) Co-expression of RSPO2 (green) and club cell marker, SCGB1A1 (red), in the isolated club cells, and RSPO2 (green) and SFTPC (red) in AT2 cells. (**F**) Immunofluorescence staining of RSPO2 (green), SCGB1A1 (red in the upper panels), and SFTPC (red in the bottom panels) in adult lung tissue section.

**Figure 2 ijms-23-03089-f002:**
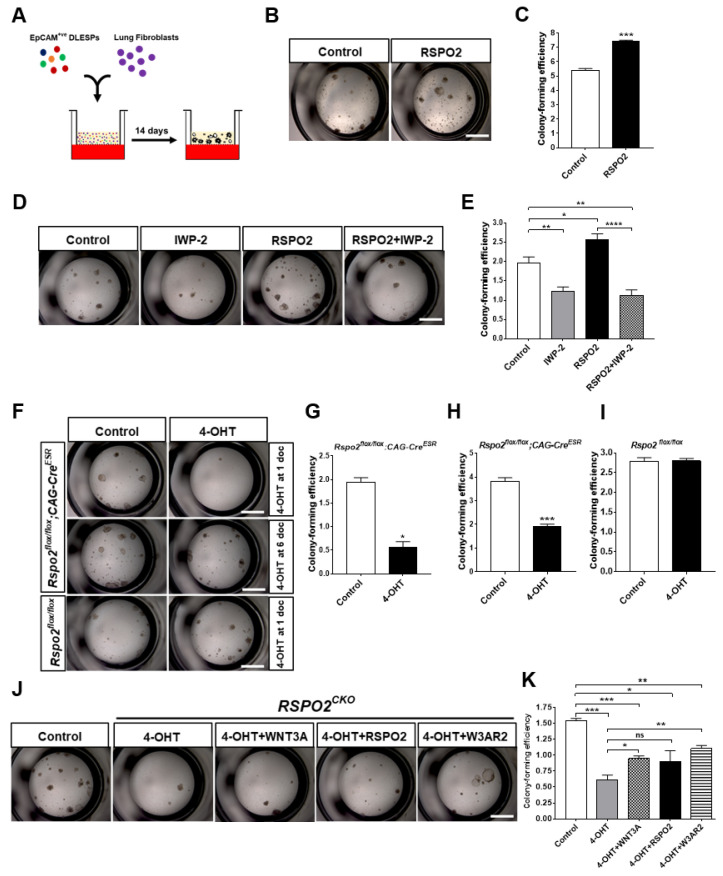
Rspo2 is essential for the colony formation of DLESPs in ex vivo 3D organoid culture. (**A**) Schematic diagram of ex vivo air–liquid interface (ALI) 3D organoid culture. (**B**,**C**) Bright-field images and quantification showing the effect of RSPO2 protein treatment (200 ng/mL) on colony formation of DLESPs in the organoid culture for 14 days. (**D**,**E**) Bright-field images and colony-forming efficiency of DLESPs showing the effect of WNT protein secretion blocker, IWP-2 (5 μM), in the organoid culture treated for 14 days. (**F**) Bright-field images of *Rspo2^CKO^* DLESP-derived organoids after 14 days of culture. Inactivation of *Rspo2* was achieved by 4-OHT (500 nM) treatment for the first two days of organoid culture. (**G**,**H**) Colony-forming efficiency of *Rspo2^CKO^* DLESPs. The cells were treated with 4-OHT (500 nM) for the first two days (**G**) or at culture day 6 for two days (**H**). (**I**) No 4-OHT effect on colony-forming efficiency of control *Rspo2^flox/flox^* DLESPs. 4-OHT treatment was administered for the first two days of the organoid culture and data were collected at day 14. (**J**,**K**) Bright-field images and colony-forming efficiency showing the rescued effects of WNT3A (20 ng/mL), RSPO2 (200 ng/mL), or W3AR2 (RSPO2 + WNT3A) protein treatment on *Rspo2^CKO^* DLESPs organoid formation. Data are presented as mean ± SEM. ns: no statistical significance, * *p* < 0.05, ** *p* < 0.01, *** *p* < 0.001, **** *p* < 0.0001.

**Figure 3 ijms-23-03089-f003:**
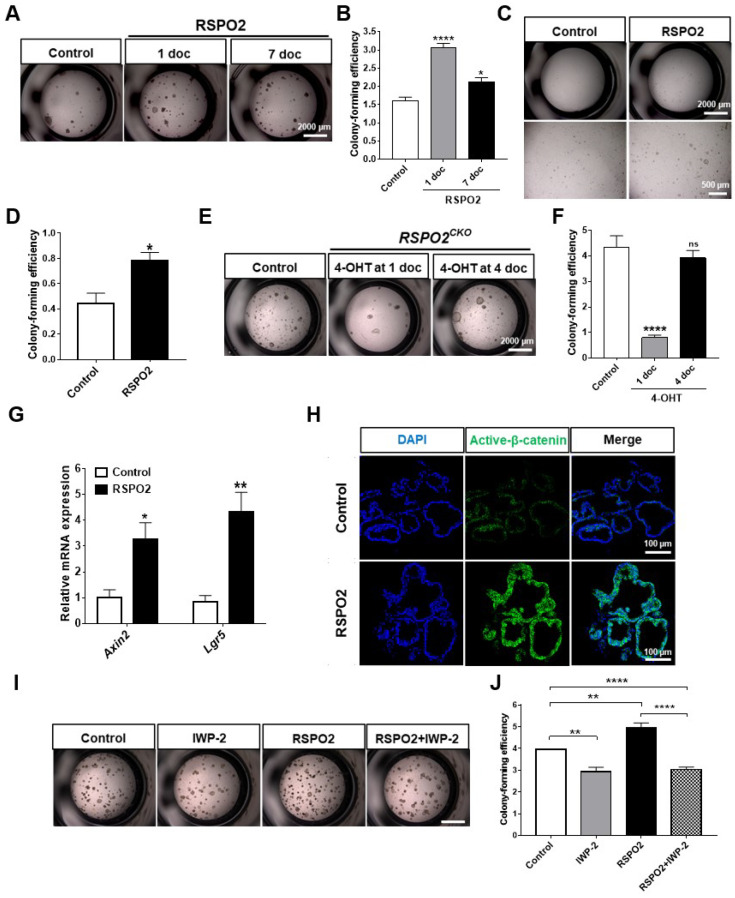
RSPO2 supports the colony formation of club cells in ex vivo 3D organoid culture through the activation of endogenous WNT/β-catenin signaling. (**A**,**B**) Bright-field images and colony-forming efficiency of club cells in the 3D organoid culture showing the effect of RSPO2 treatment (200 ng/mL). RSPO2 was administered from day 1 of culture (1 doc) or day 7 of organoid culture (7 doc). (**C**,**D**) Bright-field images and colony-forming efficiency of club cells in a fibroblast-free, FGF10/HGF supplemented 3D organoid culture. RSPO2 (200 ng/mL) was added from 1 doc for 7 days. (**E**,**F**) Bright-field images and colony-forming efficiency of *Rspo2^CKO^* club cells in the organoid culture. (**G**) RSPO2 (200 ng/mL) upregulated the expression of two WNT/β-Catenin downstream target genes, *Axin2* and *Lgr5*, in club cell-derived organoids, as determined by qRT-PCR analysis. (**H**) Immunofluorescence staining of active β-Catenin (unphosphorylated form) after RSPO2 treatment in club cell-derived organoids. Cell nuclei were counterstained with DAPI. (**I**,**J**) Bright-field images and colony-forming efficiency of club cells showing the effect of WNT protein secretion blocker, IWP-2 (5 μM), on the organoid culture (Scale bar, 2000 μm). All organoids were harvested at 7 or 14 doc for further analysis. Data are presented as mean ± SEM. ns: no statistical significance, * *p* < 0.05, ** *p* < 0.01, **** *p* < 0.0001.

**Figure 4 ijms-23-03089-f004:**
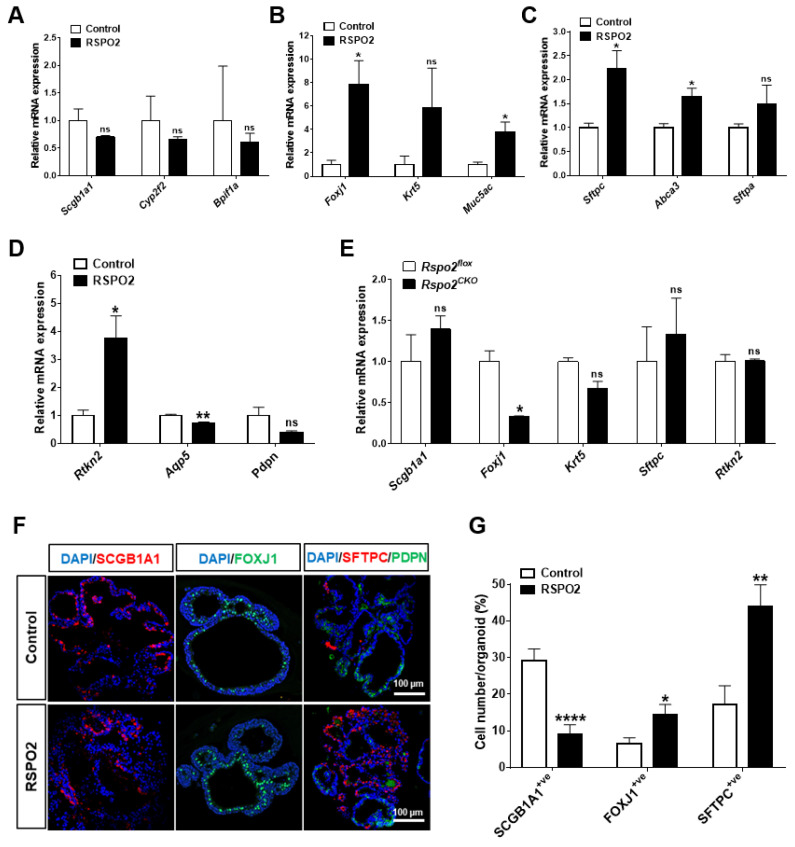
RSPO2 promotes the differentiation of club cells into multiple epithelial lineages in ex vivo 3D organoid culture. (**A**) qRT-PCR analysis of club cell marker genes, *Scgb1a1*, *Cyp2f2* and *Bpifa1*, in club cell-derived organoids treated with RSPO2 (200 ng/mL) for 14 days. (**B**) qRT-PCR analysis of gene markers for ciliated (*Foxj1*), basal (*Krt5*), and goblet (*Muc5ac*) cells in the RSPO2 (200 ng/mL)-treated club cell-derived organoids. (**C**,**D**) qRT-PCR analysis of AT2 cell marker genes (**C**) *Sftpc*, *Abca3*, and *Sftpa*, and AT1 cell marker genes (**D**) *Rtkn2*, *Aqp5*, and *Pdpn* in the RSPO2-treated club cell-derived organoids. (**E**) qRT-PCR analysis of various differentiated cell marker genes in *Rspo2^CKO^* club cell-derived organoids. (**F**,**G**) Immunofluorescence staining images for SCGB1A1, FOXJ1, SFTPC, and PDPN, and quantification of SCGB1A1^+ve^ club, FOXJ1^+ve^ ciliated, and SFTPC^+ve^ AT2 cells in the RSPO2 (200 ng/ml)-treated club cell organoids. All organoids were collected after 14 days in culture. Data are presented as mean ± SEM. ns: no statistical significance, * *p* < 0.05, ** *p* < 0.01, **** *p* < 0.0001.

**Figure 5 ijms-23-03089-f005:**
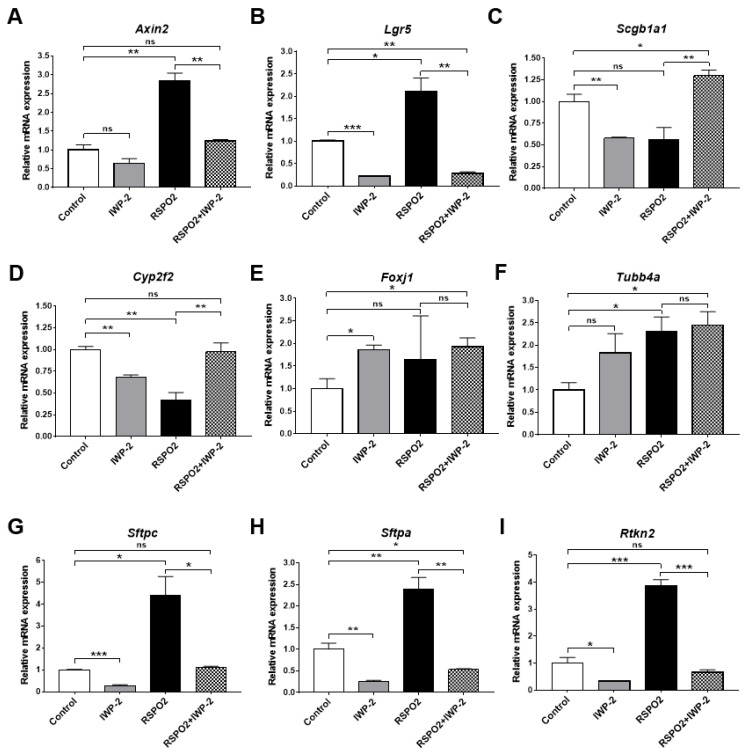
RSPO2 drives the differentiation of club cells into alveolar lineages in an endogenous WNT/β-catenin activation-dependent manner. (**A**,**B**) qRT-PCR analysis of WNT/β-catenin downstream target genes, *Axin2* and *Lgr5*, in IWP-2 (5 μM)- and/or RSPO2 (200 ng/mL)-treated club cell-derived organoids. (**C**–**I**) qRT-PCR analysis of club cell markers *Scgb1a1* (**C**) and *Cyp2f2* (**D**), ciliated cell markers *Foxj1* (**E**) and *Tubb4a* (**F**), AT2 cell markers *Sftpc* (**G**) and *Sftpa* (**H**), and AT1 cell marker *Rtkn2* (**I**) in the club cell organoids treated with IWP-2, RSPO2, or both. All organoids were collected for RNA isolation at day 14 of culture. Data are presented as mean ± SEM. ns: no statistical significance, * *p* < 0.05, ** *p* < 0.01, *** *p* < 0.001.

**Figure 6 ijms-23-03089-f006:**
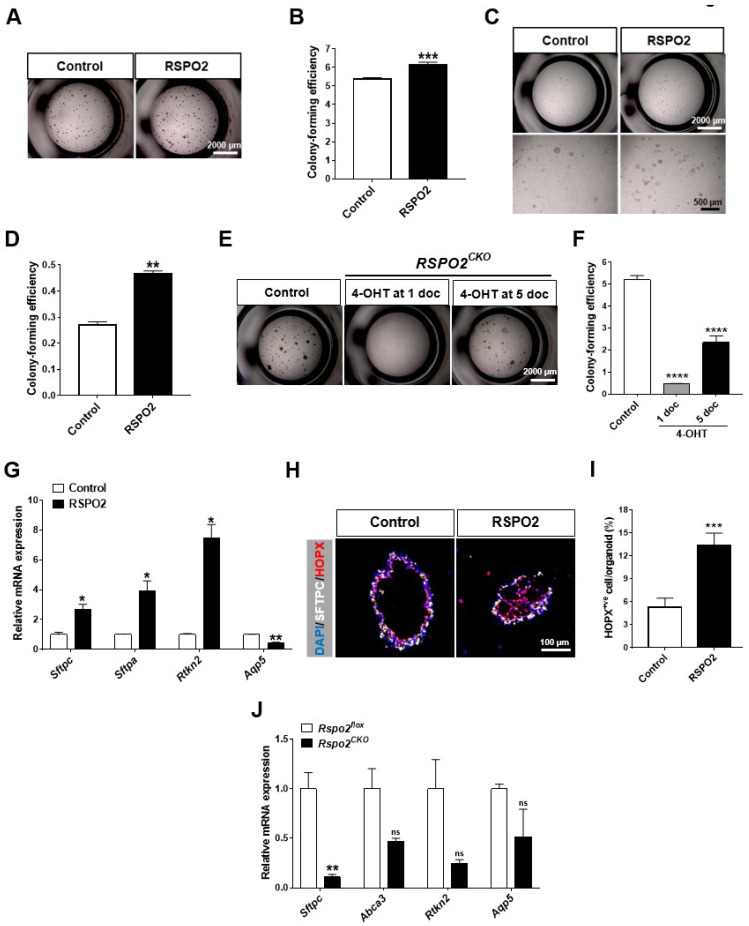
RSPO2 is essential for the colony formation and differentiation of AT2 cells in ex vivo 3D organoid culture. (**A**,**B**) Bright-field images and colony-forming efficiency of AT2 cells in ex vivo 3D organoid culture treated with in RSPO2 (200 ng/mL). (**C**,**D**) Bright-field images and quantification of colony forming efficiency of AT2 cells in a fibroblast-free, FGF10/HGF supplemented 3D organoid culture in the presence of RSPO2 (200 ng/mL). (**E**,**F**) Bright-field images and colony-forming efficiency of *Rspo2^CKO^* AT2 cells in the organoid culture. 4-OHT (500 nM) was administered for a total of two days beginning at day 1 or day 5 of organoid culture. (**G**) qRT-PCR analysis of AT2 cell markers, *Sftpc* and *Sftpa*, and AT1 cell markers, *Rtkn2* and *Aqp5*, in AT2 cell-derived organoids treated with RSPO2 (200 ng/mL). (**H**) Immunofluorescence staining of SFTPC and early AT1 cell marker, HOPX, in AT2 cell-derived organoids treated with RSPO2 (200 ng/mL). DAPI was used to counterstain cell nuclei. (**I**) Quantification of HOPX^+ve^ cells in the RSPO2-treated AT2 cell organoids. (**J**) Expression of *Sftpc*, *Abca3*, *Rtkn2*, and *Aqp5* in *Rspo2^CKO^* AT2 cell organoids by qRT-PCR. All organoids were collected at 7 doc for analysis. Data are presented as mean ± SEM. ns: no statistical significance, * *p* < 0.05, ** *p* < 0.01, *** *p* < 0.001, **** *p* < 0.0001.

**Figure 7 ijms-23-03089-f007:**
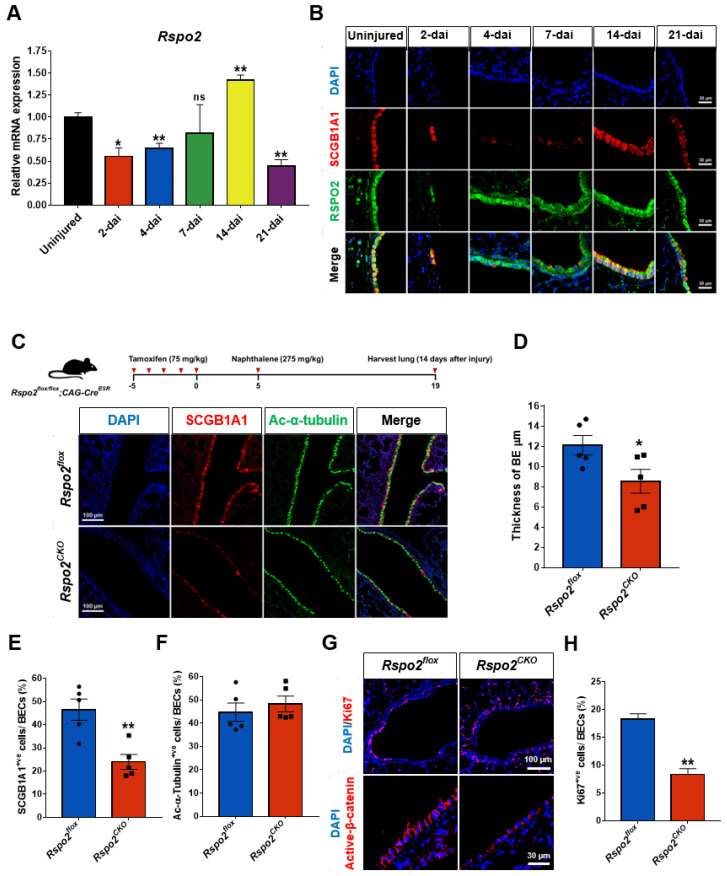
*Rspo2* gene ablation causes defects in airway regeneration after naphthalene-induced acute lung injury**.** (**A**) *Rspo2* expression during airway regeneration in naphthalene-induced lung injury by qRT-PCR. dai; days after injury. (**B**) Immunofluorescence staining of RSPO2 and SCGB1A1 during airway regeneration. (**C**) Schematic diagram of the experimental schedule for conditional *Rspo2* knockout followed by naphthalene-induced acute lung injury, and the expression of SCGB1A1 and Ac-α-tubulin determined by immunofluorescence staining of *Rspo2^CKO^* lung tissues at 14 dai. (**D**) Measurement of the thickness of the bronchiolar epithelium at 14 dai in *Rspo2^CKO^* lungs. (**E**,**F**) The percentage of SCGB1A1^+ve^ club and Ac-α-tubulin^+ve^ ciliated cells in the bronchiolar epithelium at 14 dai in *Rspo2^CKO^* lungs. (**G**,**H**) Immunofluorescence staining of Ki67 and active β-Catenin at 14 dai in *Rspo2^CKO^* lung tissue and quantification of Ki67^+ve^ proliferative cells. Data were collected from the samples (*n* = 4 for *Rspo2^CKO^* and control mice each) and are presented as mean ± SEM. ns: no statistical significance, * *p* < 0.05, ** *p* < 0.01.

## Data Availability

Materials will be available from the corresponding author upon a formal request unless there is a competing issue.

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
