# Peer review of "R-Spondin2, a Positive Canonical WNT Signaling Regulator, Controls the Expansion and Differentiation of Distal Lung Epithelial Stem/Progenitor Cells in Mice"

_ijms, 2022, doi:10.3390/ijms23063089_

Round 1
Reviewer 1 Report
The purpose of the review paper was „to determine Rspo2 function in distal lung stem/progenitor cells and adult lung regeneration”.
The work has a solid basis to be original, as it is an open question so far whether R-spondin2 (RSPO2), a co-activator of Wnt/β-catenin signaling, is expressed and plays any role in the expansion and differentiation of adult lung epithelial stem/progenitor cells during homeostasis and regeneration has not been demonstrated. The hypothesis posed in this way is interesting and the answer to this will also be important in terms of publication citations.
The authors used modern research techniques based on utilizing ex vivo ALI-organoid culture using primary stem/progenitor cells, such as DLESPs, club cells, or AT2 cells and in vivo naphthalene-induced airway injury in mice lacking Rspo2.
I have no substantive comments on the description of the research techniques. The use of italics in the descriptions from subsection 2.4 (line 165) to 2.10 (line 288) is probably unintentional. Please correct this.
The results of the paper are presented in great detail in the core material including 7 Figures and in the supplementary material (5 figures, 2 tables). All of the descriptions are very clear and not questionable, although there is a great deal of detail.
Only in the case of the description to Figure 4 A, B, C, and E - subsection 3.4. (line 436, 439, 443) (page 14) "we found that RSPO2 treatment decresed the expression of club cell markers,...please add that, however, these were not statistically significant (NS) differences, just as the increase in Krt5 was not significant, etc. ‘’.
I suggest you check the entire Figure 4 for this here and also in the discussion (line 658), as it is inaccurate. There is no increase or decrease if it is statistically insignificant.
The discussion is comprehensive, highlights the major accomplishments and results of the work, and suggests further research. It is written in a mature, professional manner and easy to read despite the large number of results obtained.
In the Discussion section - please correct that RSPO2 decreased Scgb1a1 expression, but that this was not a statistically significant difference. I also did not notice in the Figures the expression of the marker Tubb4a in club-derived organoids. Please check this as well.
Overall, the work is very valuable and fully deserves to be published.
However, please also check the publication citation requirements in parentheses as per the recommendations of the journal. So far in IJMSci, the numbers in parentheses of consecutively cited papers were applicable. Similarly, please correct (if necessary) the citations in the references list as recommended by the editor.
The study shows a previously unknown role of RSPO2 as a canonical WNT signaling regulator in adult lung epithelial stem/progenitor cells and airway regeneration.
The paper is written in very good scientific language, clearly, very understandable to the reader. The conclusions of the paper are posed correctly, I have no comments.
Author Response
We thank to reviewer 1 for very constructive suggestions and comments.
We corrected the followings according to reviewer1's comments.
- Several paragraphs written in Italics are indeed an unintentional error. We corrected them.
- In Fig 4, we modified our description to better describe the results.
- In Discussion, we also modified to better reflect the Fig4 results.
- We corrected other points in discussion as suggested.
- We used MDPI style for the references.
- We moved "materials and methods" after "discussion" to make a manuscript format recommended by IJMS.
We think our revision would address all the points raised by reviewer 1 and be good enough for publication.
Thank you for your consideration!
Best, Jeong Kyo Yoon
Reviewer 2 Report
I dont have any major comments except proofreading the paper for textual corrections and alignment of text on all pages especially results section.
Author Response
We thank to reviewer 2 for her/his time to carefully read our manuscript.
We pleased with the reviewer's comments. We formatted the manuscript appropriately to IJMS format (including the reference style).
Combining with revision suggested by reviewer 1, we think our manuscript would be good enough to be published.
Thank you for your consideration!
Best,
Jeong Kyo Yoon